# The Efficacy of Molecular Analysis in the Diagnosis of Bone and Soft Tissue Sarcoma: A 15-Year Mono-Institutional Study

**DOI:** 10.3390/ijms24010632

**Published:** 2022-12-30

**Authors:** Stefania Benini, Gabriella Gamberi, Stefania Cocchi, Giovanna Magagnoli, Angela Rosa Fortunato, Enrica Sciulli, Alberto Righi, Marco Gambarotti

**Affiliations:** 1Department of Pathology, IRCCS Istituto Ortopedico Rizzoli, 40136 Bologna, Italy; 2Department of Biomedical and Neuromotor Sciences, University of Bologna, 40126 Bologna, Italy

**Keywords:** molecular diagnostics, fusion transcript, sarcomas, frozen tissue, formalin-fixed, paraffin-embedded tissue, RT-PCR, qRT-PCR, FISH, next-generation sequencing

## Abstract

The histological diagnosis of sarcoma can be difficult as it sometimes requires the combination of morphological and immunophenotypic analyses with molecular tests. A total of 2705 tissue samples of sarcoma consecutively collected from 2006 until 2020 that had undergone molecular analysis were assessed to evaluate their diagnostic utility compared with histological assessments. A total of 3051 molecular analyses were performed, including 1484 gene fusions tested by c/qRT–PCR, 992 gene rearrangements analysed by FISH, 433 analyses of the gene status of MDM2, 126 mutational analyses and 16 NGS analysis. Of the samples analysed, 68% were from formalin-fixed, paraffin-embedded tissue and 32% were from frozen tissue. C/qRT–PCR and FISH analyses were conclusive on formalin-fixed, paraffin-embedded tissue in 74% and 76% of samples, respectively, but the combination of the two methods gave us conclusive results in 96% and 89% of frozen and formalin-fixed, paraffin-embedded tissues, respectively. We demonstrate the utility of c/qRT–PCR and FISH for sarcoma diagnosis and that each has advantages in specific contexts. We conclude that it is possible to accurately predict the sarcoma subtype using a panel of different subtype-specific FISH probes and c/qRT–PCR assays, thereby greatly facilitating the differential diagnosis of these tumours.

## 1. Introduction

Sarcomas are a heterogeneous group of mesenchymal neoplasms for which histological assessment is challenging. The morphology and immunoprofile of different tumour types frequently overlap and can mimic other tumour types. Nevertheless, an accurate diagnosis of many of these tumours remains a challenge, since they often share similar morphological features and similar cellular differentiation [1,2,3]. Of importance, numerous recurrent chromosomal abnormalities, especially balanced translocations, have been associated with different subtypes of bone and soft tissue sarcomas (BSTSs). Thus, the recognition of these abnormalities could be exploited for the differential diagnosis of these subtypes [4,5,6].

The diagnosis of BSTSs is challenging as their morphology and immunophenotype often overlap with each other and with tumours of other lineages, such as carcinomas and melanomas. Many BSTSs are associated with characteristic, reproducible genetic aberrations (chromosomal translocations that generate novel chimeric fusion genes, gene amplifications and gene mutations), which can be used for diagnostic purposes. As most genetic aberrations are tumour-specific, their detection by cytogenetic or molecular genetic techniques is diagnostically useful. Translocations can be detected by fluorescence in situ hybridization (FISH), and novel fusion transcripts can be detected by conventional or quantitative reverse-transcriptase polymerase chain reaction (c/qRT–PCR). The ability to achieve reliable molecular test results from formalin-fixed, paraffin-embedded (FFPE) tissue has been well documented [7,8,9]. Fresh frozen tissue remains optimal but attaining such material in routine histopathological practice is generally impractical, particularly for consultation cases.

The aim of this study, therefore, was to evaluate the impact and feasibility of the analysis of chromosomal alterations using interphase FISH and c/qRT–PCR in BSTSs, without a known diagnosis at the time of investigation. To this end, we reassessed 15 years of activity by comparing the use of FFPE tissue vs. the frozen tissue to see whether there were discernible improvements in the efficacy of rearrangement and *MDM2* gene amplification tests. In the last 5 years, routine molecular testing for giant cell tumour (GCT) has been performed to assess the mutation status of *H3F3A*, and molecular testing for chondroblastoma using the *H3F3B* gene status has also been introduced. Experience with these techniques has also increased within the department as more molecular tests are routinely performed. In recent years, the use of a novel technique, anchored multiplex PCR (AMP) for next-generation sequencing (NGS) using the Archer Fusion Plex Sarcoma kit, has been implemented. Recently, assays have been established for fusion gene transcript detection that combine multiplex anchored PCR amplification of specific targets with NGS [10,11]. This study also evaluated the frequency of the samples that needed a thorough investigation by NGS to evaluate the diagnostic impact and feasibility of NGS.

## 2. Results

### 2.1. Molecular Methodology

In the present study, we collected the results of molecular analyses that were performed over 15 years using different c/qRT–PCR and FISH assays. All cases with a request of molecular investigation for fusion transcripts, gene rearrangement analysis, *MDM2* amplification and gene variations were included in this study. In the case of molecular investigation for fusion transcript, the workflow followed the algorithm, as shown in Appendix A. Most of the PCR assays have been used according to the available scientific literature [12,13,14,15,16]. Additionally, some have been designed and validated in our laboratory, and they were designed exclusively for molecular investigations on FFPE tissue (Appendix A).

The tumours most frequently tested for molecular confirmation were Ewing sarcoma (ES) (761 patients), well/dedifferentiated liposarcoma (*MDM2* status) (396 patients), synovial sarcoma (SS) (393 patients), myxoid/round cell liposarcoma (MLPS) (388 patients) and extraskeletal myxoid chondrosarcoma (EMC) (125 patients). Less frequent tests were performed for desmoplastic small round cell tumours (nine patients), low-grade fibro myxoid sarcoma (LGFMS) (nine patients), sclerosing epithelioid fibrosarcoma (nine patients), angiomatoid fibrous histiocytoma (eight patients) and pseudomagnetic haemangioendothelioma (three patients). The summaries of the findings for each tumour type tested are shown in Table 1.

### 2.2. Molecular Results

Molecular tests were requested for material from 2705 bone and soft tissue lesions. Of these tissues, 68% (1903) consisted of FFPE tissue samples (590 were from blocks prepared elsewhere for a second opinion or for diagnostic review) and 32% (884) were from processed frozen tissue from our institution (Figure 1A). In total, 3051 analyses were carried out. Of these analyses, 1610 were c/qRT–PCR, 1484 were fusion transcript analyses performed on mRNA samples and 126 were gene variant analyses on DNA samples. The total number of FISH analyses was 1425. Of these analyses, 992 were gene rearrangement analyses and 433 were evaluation of the *MDM2* gene status (Figure 1B). In 14% of the gene rearrangement analyses, both methods were used to assess a specific tumour type (including those having multiple tests for differential diagnoses), and in the remaining cases, only one method was used (Figure 1C).

Of the c/qRT–PCR analyses performed on frozen tissue samples, 91% were evaluable, 1% were not evaluable and 8% were unsuitable material. Of the c/qRT–PCR analyses performed on FFPE tissue samples, 74% of cases were evaluable, 10% were not evaluable and in 16% the material was unsuitable for analysis (Figure 2A). Similarly, the FISH analyses performed on frozen and FFPE tissues were evaluable in 91% and 76% of cases, respectively, and not evaluable in 7% and 9% of cases, respectively. Unsuitable material was observed in 2% of cases of FISH performed on frozen tissue samples compared with 15% of cases of FISH performed on FFPE tissue samples (Figure 2B). Of the unsuccessful c/qRT–PCR, 33% were consultation cases (reference samples) and 25% were decalcified tissue. For the FISH tests, 45% were reference samples and 36% were decalcified tissue. The combination of the two methods of analysis made it possible to obtain an evaluable response in 96% of the frozen samples and 89% of FFPE tissue samples. However, a non-evaluable response was obtained in 4%, and an unsuitable response was obtained in 7% (Figure 2C). The gene variant analysis performed on DNA extracted from frozen and FFPE tissues was evaluable in 94% and 87% of cases, respectively. It was not evaluable in only 2% of the FFPE tissue samples, and unsuitable material was observed in 6% and 11% of frozen and FFPE tissues samples, respectively (Figure 3D). The evaluation of gene rearrangements in the most frequently analysed tumours showed that c/qRT–PCR was more sensitive than FISH in detecting ES and SS (94% and 93%, respectively, vs. 85% and 75%), but FISH was more sensitive than c/qRT–PCR in detecting MLPS and EMC (99% and 94%, respectively, vs. 92% and 88%) (Table 2).

To evaluate the analytical quality of the samples tested for the amplification of the *MDM2* gene with the FISH method, the samples transported in formalin solution were compared with the samples transported under vacuum conditions. In the latter case, the samples were sealed in plastic bags in a vacuum apparatus immediately after surgical removal (Tissue-safe®, Milestone, Bergamo) [17]. Samples stored and transported with the use of formalin solution were evaluable in 67% of the samples, not evaluable in 14% of the samples, and unsuitable in 19% of the samples. The samples immediately stored and transported under vacuum were evaluable in 84% of the samples, not evaluable in 5% of the samples and unsuitable in 11% of the samples (Figure 3). Statistical analysis between the group with and without the use of Tissue-safe shows that these difference were statistically significant (*p* < 0.0001).

### 2.3. NGS Results

NGS was performed for 16 cases that were determined to be negative by the first molecular screening (c/qRT–PCR and FISH); they were received for diagnostics (in the period 2019–2020). Of the samples analysed, nine cases (57%) did not have fusions, and six (37%) cases had fusions. One sample (6%) failed due to poor material quality and was determined to be unsuitable by failing the minimum quality control for parallel massive sequence analysis (Appendix A). Regarding the six cases with fusions, Case 1 resulted in a new rearrangement, EWSR1-VEZF1, that was previously described [18] and was confirmed by qRT–PCR with primers designed based on the breakpoint sequence. In cases 4 and 15, characteristic ES gene fusions were detected (EWSR1-FLI1 in case 4 and EWSR1-FEV in case 15). Case 4 was a previous patient-requested NGS analysis that was confirmed by qRT–PCR. Case 15 tested negative on c/qRT–PCR at the most frequent translocations for ES, while the *EWSR1* analysis in FISH had a very doubtful setting, with split signals observed in 28 of 200 (14%) nuclei. A repetition of FISH on the same paraffin block showed similar results. Targeted NGS showed a translocation involving *EWSR1* and *FEV*; the presence of a translocation was confirmed by cRT-PCR. For Case 5, FISH indicated that *EWSR1* was rearranged, and NGS subsequently revealed a fusion between *EWSR1* and *PBX3* (Figure 4A,B). Gene fusions in *PBX3* are not commonly described in myoepithelial tumours. Using specially designed primers that were based on the target NGS-derived sequences of the fusion transcript and covered the breakpoint, cRT-PCR revealed the correct fusion product. This result was subsequently confirmed with Sanger sequencing (Figure 4C). Case 10 tested negative for ES and Ewing-like tumours (*CIC* sarcoma and *NFATC2*-reaarranget tumour), but targeted NGS revealed a fusion between *EWSR1* and *ATF1*. This result was confirmed by qRT–PCR. Case 12 tested negative in the MLPS assay by c/qRT–PCR and FISH. Subsequently, NGS revealed a fusion between SS18 and SSX1, which was confirmed by a qRT–PCR assay (Appendix A).

Clinically relevant molecular analysis requires results to be reported in a clinically useful timeframe. We tracked the laboratory turnaround time (TAT) by retrieving specimen status data from the laboratory information system. Figure 5A shows the key steps in the workflow of a generic molecular analysis, culminating in reporting the patient’s electronic medical record. The median preanalytical time, including delays related to obtaining tissue from outside institutions, was 7 days (range, 1–63 days; Figure 5B). The median TAT in the molecular pathology laboratory from the start of the RNA/DNA extraction to sign out, was 3 days (range, 1–24 days; Figure 5B). Although some cases required orthogonal (Sanger) confirmation of certain variants, this was not a major determinant of TAT. Although RT–PCR and FISH tests have been found, in our experience, to be relatively rapid, the platform used for sequencing was a much more meaningful determinant of TAT (Figure 5B).

## 3. Discussion

The detection of tumour-specific genetic aberrations is becoming an integral part of the diagnostic investigation of bone and soft tissue tumours [19,20,21,22,23]. Given the increasing spectrum of rearranged neoplasms, it is clear that the utility and practicality of ancillary molecular tests need re-evaluation. Furthermore, it is important that physicians are aware of the limitations of molecular diagnostic techniques, including the nonspecificity of many gene fusions. The two main diagnostic platforms available to routine surgical pathology laboratories are FISH and c/qRT–PCR, which are most beneficial when used as complementary modalities. The spectrum of gene fusions associated with BSTS is far wider than previously anticipated.

This study reports all molecular diagnostics performed at the Rizzoli Pathology Department; 2705 samples (68% from FFPE tissue and 32% from frozen tissue) of suspected tumours of the musculoskeletal system that were collected over 15 years were analysed. Specific molecular abnormalities were identified in 1669 of 2705 cases (62%) of sarcomas, including 693 of 986 bone tissue sarcomas (70%). A total of 1610 investigations were carried out by c/qRT–PCR using a panel of 63 different c/qRT–PCR assays developed for fusion transcript analysis of FFPE and frozen tissues from 22 different tumour types. To the best of our knowledge, this is the largest analysis of sarcomas based on molecular studies. Based on the availability of the material, the first analysis would usually first involve the c/qRT–PCR and subsequent FISH analysis. Only the FISH method was needed for *MDM2* analysis or when the material was limited (e.g., in a biopsy sample with few tumour cells, the FISH analysis requires only one section of 3 µm compared to 2 × 10 µm for c/qRT–PCR). In our experience, the c/qRT–PCR was the more frequently used ancillary test for gene rearrangement because it can quickly provide an answer, as demonstrated by the evaluation of TAT. Another reason for the prevalence of c/qRT–PCR tests is related to the availability, in our laboratory, of a large panel of assays (Appendix A) for gene fusion transcripts.

In determining the sensitivity and specificity of the molecular detection, here we report that c/qRT–PCR tests have greater sensitivity and specificity than FISH analysis in the tumours most frequently tested (ES and SS). These results are in agreement with a previous paper reporting that FISH and RT-PCR show similar sensitivities, while RT-PCR showed a higher specificity [24]. Whereas our algorithm first provides c/qRT–PCR followed by FISH in ES and SS tumours, in the case of myxoid liposarcoma and extraskeletal myxoid chondrosarcoma, the algorithm is reversed and FISH analysis was found to be more sensitive than c/qRT–PCR. As previously described, FISH approaches, using specific translocation probes, are highly sensitive and specific for the molecular diagnosis [25]. However, in some tumours, RT–PCR shows equal sensitivity and may detect cases that are determined to be negative by FISH [24,26].

In this study, we evaluated the impact of preanalytical factors and the quality control measures required to improve the reliability of RNA extraction from FFPE material. RNA expression analysis from FFPE tissues has been broadly used in biomedical research [19,27]. It is known that RNA extracted from FFPE tissue is degraded into smaller fragments than those from frozen tissue [28,29]. All the frozen samples analysed in this study were internal from the institute, while the 32% of the FFPE tissues came from external consultations. The analyses performed from FFPE tissue were evaluable in 74% and 76% of c/qRT–PCR and FISH tests, respectively. In contrast to FFPE tissue, the analyses performed on frozen tissue were evaluable, for both methods, on 91% of tests performed. In cases where it was possible to combine the two methods, we obtained evaluable results in 89% of cases reducing the percentage of failures of the analyses performed on FFPE tissues. As previously described, comparisons between FISH and RT-PCR are complementary and suggest that the combination of both methods is recommended for the diagnosis of BSTS [24,30]. The same trend was also observed for the gene variant analyses performed on DNA extracted from frozen and FFPE tissues, which had reliabilities of 94% and 87%, respectively. As expected, these results confirm the fact that the success rate in the evaluation of gene fusions and gene variants in frozen tissue are much higher than that in the FFPE tissue. Although the literature reports that RT–PCR has a higher failure rate than FISH (described in EWSR1-rearranged neoplasm) [23], in this study the failure rates of the two methods are comparable.

This study shows a failure rate of 26% in the c/qRT–PCR performed on FFPE material, including unsuitable/not amplifiable material (16%) and uninterpretable results (10%). The RNA is prone to degradation, and the efficiency of extracting amplifiable nucleic acids from FFPE tissue is dependent on the calcification, ischemia and fixation time [29,31]. Despite the use of FISH being proven to have a lower failure rate than RT-PCR [14,23], in this study we report that the FISH investigations have a failure rate of 25%, similar to c/qRT–PCR, including unsuitable material (15%) and uninterpretable results (9%). Several preanalytical parameters, such as ischaemia times, type of fixative, duration of fixation, storage and extraction techniques of biomolecules, impact the quality of RNA and DNA from FFPE tissues. To overcome this issue, several improvements to the standard FFPE protocol have been proposed, such as fixation at a low temperature or vacuum storage [32]. The impact of multiple preanalytical factors, including time from excision to fixation, the volume of tissue to the volume of fixative, duration of fixation, postprocessing factors, and the thickness of the tissue specimen cut onto the glass microscope slide, were critical for optimal molecular analysis performance. The control of fixation time and temperature is recommended for molecular analyses. The vacuum packing of tissue and transfer to the laboratory at 4 °C (Tissue-safe) is one option attracting attention and can be very useful for sites distant to the hospital laboratory [33]. Cooling should be as rapid as possible, and cold fixation preserves RNA well. However, cold shock-induced changes should be a consideration for some biomarkers [32]. The introduction of controlled temperature in the vacuum tissue transport system has allowed us to improve the performance of molecular diagnostic tests, as observed in the *MDM2* gene amplification test. It is probably optimal for the fixation process to be controlled by the pathology laboratory. This may involve the transport of fresh specimens to the pathology laboratory, preferably under vacuum for transport times of more than an hour.

This study we also report our experience with recently implemented NGS technology used for all cases that have not been diagnosed with c/qRT–PCR and FISH. A total of 100% and 90% of samples were evaluable from fresh and FFPE tissues, respectively. Of the samples tested, 57% did not have gene fusions, while we found the presence of gene fusions in 37%. Only 6% (one case from FFPE tissue) did not pass the PreSeqRNA quality control; these results are in agreement with those reported before [14]. Regarding the samples with fusions, only two cases had a revised diagnosis (Case 10 with EWSR1/ATF1 and Case 12 with SS18/SSX); for both cases the morphology and immunohistochemical results were questionable. However, for the other cases, the NGS results gave molecular confirmation of an already supposed diagnosis. This study and those of others [8,14] show that NGS can be performed on FFPE tissue, and it is expected that these studies will soon expand. NGS technologies continue to technically improve and become cheaper and easier to use. Thus, it is possible that NGS may gradually replace outdated methods for detecting gene fusions, not only in research projects but also in the clinical setting. Hopefully, this will be facilitated by the development of more sophisticated algorithms for data analysis [34].

At present, in our experience, the methodological approach to evaluate gene fusions, which first involve c/qRT–PCR analysis and then FISH, has proven to be rapid, economical and reliable. TAT is a key aspect of clinical feasibility but has not previously been well documented for a clinical NGS cancer test. The combination of PCR and FISH allowed us to diagnose 54% of the samples in 3 days and 95% in 8 working days. However, a realistic increase in TAT, was observed for the samples that required an in-depth diagnosis by NGS. Some samples required multiple attempts at tissue and RNA preparation. Some samples required FISH and/or RT–PCR for orthogonal confirmation. Some samples showed fusion that required laborious interpretation and preparation and development of a specific c/qRT–PCR assay and performing Sanger sequencing for confirmation.

In conclusion, using the two methods ensures a higher success rate as both have limitations. RT–PCR is a rapid test designed to detect specific fusion transcripts, but this also means that rare fusion transcripts, or unusual translocation breakpoints, cannot be detected. Conversely, FISH assays utilize ‘break-apart’ probes that only identify a breakpoint in one of the common genes (e.g., *EWSR1* present in different tumours [23,35]) without providing information on the translocation partner. Therefore, using both methodologies provides the higher sensitivity and specificity for the detection of fusion genes in samples [19]. Therefore, all hospitals should have defined referral pathways for the diagnosis of putative soft tissue tumours, with all cases being amenable to facilities for ancillary molecular diagnosis [36,37,38]. The increased application of fusion evaluation techniques will likely lead to the discovery of numerous novel gene fusions and improve our understanding of the occurrence of known fusions. The detection of larger numbers of specific fusion tumours will broaden our understanding of the clinicopathologic correlation of various fusion tumours.

## 4. Materials and Methods

### 4.1. Sample Collection

A series of 2705 tissue samples consecutively collected from January 2006 until December 2020 that had undergone FISH, c/qRT–PCR, gene mutational analysis or next-generation sequencing (NGS) were performed. The study was approved by the ethical institutional committee (code: TRASLOCASARC CE AVEC: 729/2020/Oss/IOR).

The material assessed was that of tumours biopsied or excised at the Rizzoli Institute and processed locally, as well as tumours in paraffin blocks from other institutions or sent for a second opinion on a diagnosis. Since January 2017, a new method was implemented to improve the transport of samples in our institution. Briefly, the tissue samples were vacuum sealed in plastic bags using a semi-professional machine (Mod. VAC 10, by Milestone, Bergamo, Italy) inside the surgical theatre immediately after surgery and kept at 4 °C until transfer to the pathology laboratory. Once in pathology laboratory, the surgical specimens were processed without delay.

Tests were requested to confirm or exclude a histological diagnosis. Tumours with an obvious histological diagnosis (e.g., biphasic synovial sarcoma) were not analysed by molecular tests, and most cases tested were those with a degree of diagnostic uncertainty. All small round cell sarcomas with molecular tests performed on FFPE tissue and fresh material were included. Tests were requested to confirm or exclude a histological diagnosis. Tumours and tests for which routine molecular diagnostic assays are available in the laboratory are listed and described in Appendix A.

All cases with a request of molecular investigation of fusion transcripts, gene rearrangement analysis, *MDM2* amplification and gene variations were included in this study. In the case of molecular investigation for fusion transcript, the workflow followed the algorithm, as shown in Appendix A. According to the pathologist’s request and based on the availability of the material, the first step involved molecular investigation with c/qRT–PCR analysis; all positive results produced a molecular report. The undiagnosed cases were subsequently analysed by FISH. Based on the result (rearranged/not rearranged gene, unsuitable material or unsuitable result), the molecular report was produced and filled in. In recent years (since 2019), NGS technology has been implemented in diagnostics for all cases that have not been diagnosed with c/qRT–PCR and FISH. It has only been possible to carry out a molecular study using NGS technology in selected cases. Most cases for which molecular analysis was performed carried a degree of uncertainty as tests were carried out either for confirmation of a suspected entity or for exclusion of a possible, but less likely, tumour type.

### 4.2. Molecular Testing

For analysis of fusion transcripts, total RNA was isolated from frozen or FFPE tissue using a modified method, including TRIzol reagent (Invitrogen, Carlsbad, CA, USA) and RNeasy Mini Kit spin columns (Qiagen GmbH, Hilden, Germany), as previously described [12]. To extract RNA from FFPE tissue, sections 6–8 μm in thickness were cut from the representative paraffin block. To extract RNA from frozen tissue, a fragment of less than 10 mg of tumour tissue was selected using scalpel blades. Haematoxylin–eosin cryostat sections were performed to evaluate if the area of tissue selected was representative of the tumour. The RNA (0.2 μg to 1 μg) was then reverse transcribed to cDNA using VILO IV Master Mix (Applied Biosystems, Foster City, CA, USA). Negative controls (RNA from normal tissue and no template control) and positive controls were included in each c/qRT-PCR reaction. C/qRT-PCR was performed to assess for fusion transcripts according to methods described in Appendix A. Briefly, 63 different c/qRT-PCR assays were developed of fusion transcripts analysis on FFPE and frozen tissue on 22 different tumour entities. In addition, 6 assays were validated for the analysis of gene variants detectable both on frozen and FFPE tissue by PCR and Sanger sequencing. Of all the PCR assays used, 12 were designed and validated by our laboratory and published for the first time in this study (Appendix A), 7 are commercial (Appendix A) and the remaining 50 are from the literature (Appendix A). For gene variant analysis, total DNA extraction from FFPE tissue samples was performed using the QIAamp FFPE Tissue kit (Qiagen, Valencia, CA, USA), according to the manufacturer’s instructions.

With some modifications the same QIAamp FFPE Tissue kit (Qiagen) was used for DNA purification of frozen samples. A fragment of less than 10 mg tumour tissue was selected using scalpel blades. Haematoxylin–eosin cryostat sections were performed to evaluate if the area of tissue selected was representative of the tumour. For PCR reactions, >50 ng of DNA was amplified using AmpliTaq Gold 360 Master Mix (Applied Biosystems, Foster City, CA, USA), according to described methods, with 0.5 μM of primers (Appendix A). Reaction controls included the no template control, wild-type control and mutated positive controls. The sequencing was performed by Bio-Research Fab (Rome, Italy). Mutation analysis was conducted with Basic Local Alignment Search Tool (BLAST) in the NCBI database “National Center of Biotechnology Information Database”. Electropherograms were exported to FASTA format and were aligned to the NCBI BLAST sequences for each analysed gene. For each patient, results of c/qRT-PCR and FISH were compared, and these findings were compared with the histological analysis.

### 4.3. FISH Analysis

For FISH, 2–4 mm thick FFPE sections were treated, as previously described. FISH was performed using specific Dual Color (Break-apart) DNA probes, as described in Appendix A. FISH was performed using the Histology FISH accessory kit (Dako, Glostrup, Denmark), according to the manufacturers’ protocol. In brief, three-micron-thick FFPE tissue sections were mounted on positively charged slides. A haematoxylin–eosin-stained section from each tumour was prepared, and areas of representative non-necrotic neoplasm were marked by the pathologist. Slides were heated overnight (60 °C), deparaffinized with xylene and dehydrated with ethanol. Samples and probes were co-denaturated in a Dako Hybridizer (Dako, Glostrup, Denmark) at 75 °C for 10 min and incubated overnight at 37 °C. Slides were then washed in stringent solution for 10 min at 63 °C and stained with DAPI (Vector Laboratories, Inc. Burlingame CA, USA). A minimum of 100 tumour cell nuclei with intact morphology, as determined by DAPI counterstaining, were counted in the pre-marked neoplastic area. Nuclei with no signal and signals in the overlapped nuclei were considered noninformative and not analysed to avoid truncation or overlapping artifact. For gene rearrangement evaluation, a fused or closely approximated green–red signal pattern was interpreted as a normal result, whereas a split signal (break-apart) pattern (separation of red and green signals > 3 signal diameters) indicated the presence of a gene rearrangement. Positive score was interpreted when at least 10% of nuclei showed a break-apart pattern. For evaluating the status of *MDM2* gene, cases were scored by counting a minimum of 100 tumour cell nuclei at X100 magnification with a DAPI/green/red triple band pass filter. The number of *MDM2* and CEP12 signals was determined and a *MDM2*/CEP12 ratio was calculated for each nucleus. A ratio >2.0 in at least 10% of nuclei was considered amplified for the *MDM2* gene. A Color View III CCD camera soft imaging system (Olympus) was used to capture images, then analysed with a CytoVision imaging software version 7.5 (Leica Biosystem Richmond Inc, Richmond, IL, USA).

### 4.4. Anchored Multiplex PCR (AMP)-Based Targeted NGS

The NGS technology has been introduced in diagnostics since 2019. The measurement of RNA quantity for both FFPE and frozen material was measured by Qubit RNA HS Assay Kit (Thermo Fisher Scientific, MA, USA), in accordance with Archer FusionPlex Protocol for Ion Torrent. This test relies on Anchored Multiplex PCR (AMP™) technology to generate target-enriched libraries. The methodological details are described elsewhere [13,14,15]. As described previously [16], cDNA libraries for sequencing studies were prepared utilizing the Archer FusionPlex^®^ Sarcoma Panel kit (ArcherDX, Boulder, CO, FusionPlex^®^ Sarcoma and FusionPlex^®^ Sarcoma v2) following the manufacturer’s protocol. Archer Library preparation reagents included FusionPlex Reagent and unidirectional gene-specific primer (GSP1 and GSP2) targeting 26 (FusionPlex^®^ Sarcoma) or 63 (FusionPlex^®^ Sarcoma v2) genes involved in bone and soft tissue tumours and Archer MBC adapters for Ion Torrent (Archer, Boulder, CO, USA). This test relies on Anchored Multiplex PCR (AMP™) technology to generate target-enriched libraries. To assess the initial RNA quality, the Archer PreSeqRNA quality control (QC) assay was performed. If satisfactory for testing (PreSeq qPCR crossing point 30), cDNA libraries were produced. During library preparation, the cDNA was purified using Agencourt AMPure XP beads (Beckman Coulter, CA, USA). Final libraries were quantified with the Ion Library TaqMan Quantitation Kit (Thermo Fisher Scientific, MA, USA) and pooled to equimolar concentration. The prepared libraries were loaded on an Ion 520 or 510 chip using the Ion Chef system (Thermo Fisher Scientific, MA, USA) and sequenced using the Ion Torrent Personal Genomic Machine (PGM) or S5 Machine. With the Archer analysis software version 6.0.4 (ArcherDX), the produced libraries were analysed for presence of relevant fusions. The sequence quality was assessed by the following criteria: a minimal total read number of 1.5 million with >7% unique fragments, and >40% RNA reads. If all criteria were met, the specimen was considered negative for fusions in the selectively captured genes. Archer Analysis Software version 6.0.4 at default parameters was used for reads alignment, fusion genes identification, data visualization and annotation. Human reference genome hg19/GRCh37 was used for comparison to obtained sequences.

## Figures and Tables

**Figure 1 ijms-24-00632-f001:**
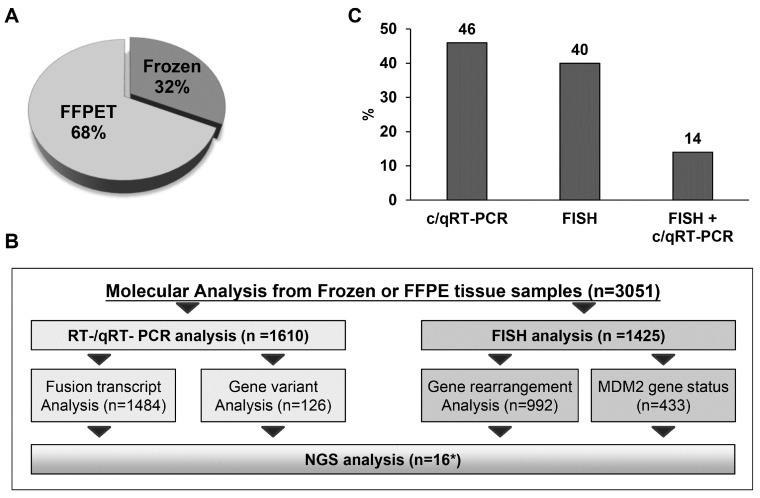
Samples and molecular analyses carried out in our institution in 15 years of activity. (**A**) The diagram shows the distribution of the type of material analysed in this study. (**B**) Schematic representation of all molecular analyses carried out in our institution in fifteen years of activity. (**C**) The diagram shows the percentage of molecular investigations performed, based on type of analysis to evaluate the presence of fusion transcript, gene rearrangements, gene amplification and gene variation. (* for NGS the number corresponds to two years of activity).

**Figure 2 ijms-24-00632-f002:**
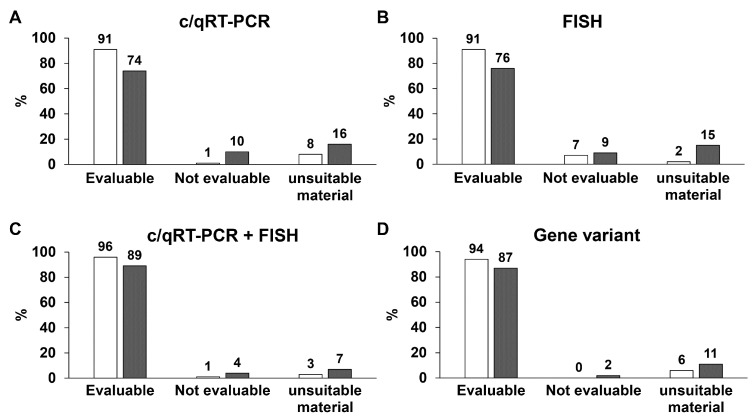
Evaluation of the analytical quality of the samples under examination comparing frozen tissue vs. FFPE tissue, based on the type of analysis. (**A**) Analytical quality of c/qRT-PCR tests on frozen tissue samples vs. FFPE tissue. (**B**) Analytical quality of FISH tests on frozen tissue samples vs. FFPE tissue. (**C**) Analytical quality of combination of the two methods c/qRT-PCR + FISH tests on frozen tissue samples vs. FFPE tissue. (**D**) Analytical quality of gene variant tests on frozen tissue samples vs. FFPE tissue. White bars represent frozen tissue samples, and grey bars represent FFPE tissue samples.

**Figure 3 ijms-24-00632-f003:**
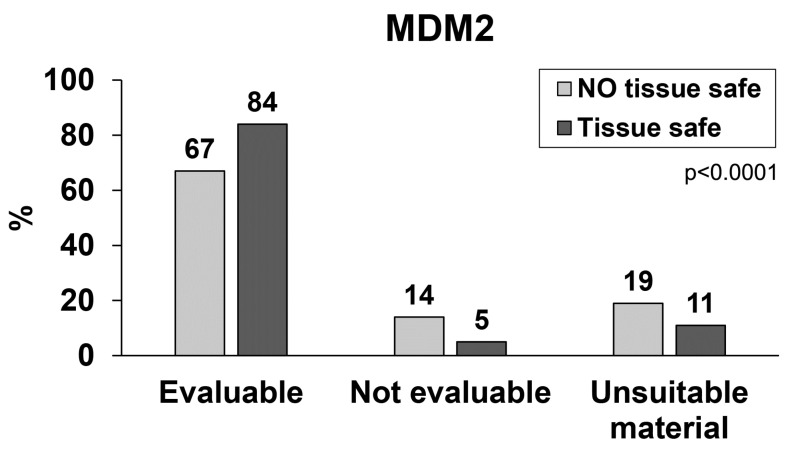
Evaluation of the analytical quality of samples analysed for the state of the *MDM2* gene by FISH on tissue stored and transported in a traditional way involving the use of formalin solution (light grey bars) vs. sealing tissue specimen under vacuum in plastic bags (dark grey bars). *p* value refers to the differences in the analytical quality (evaluable, not evaluable and unsuitable material) between the two groups.

**Figure 4 ijms-24-00632-f004:**
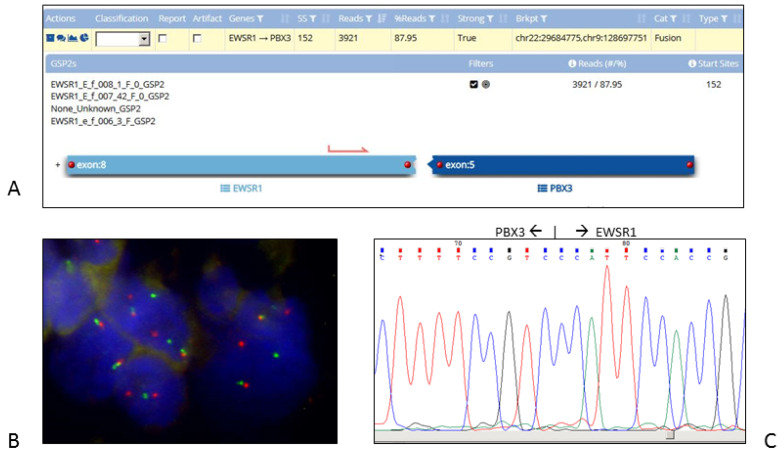
Identification of the EWSR1-PBX3 fusion transcript in case 5 (malignant myoepithelioma). (**A**) Analysis of anchored multiplex PCR result of the Archer FusionPlex Sarcoma Panel showed EWSR1-PBX3 fusion in the biopsy. (**B**) FISH of interphase nuclei using dual-colour break-apart probe for *EWSR1* (22q12) gene reveals 1 non rearranged orange/green fusion signal, 1 separate orange signal and 1 separate green signal, indicating the translocation of *EWSR1* in tumour cells. Original magnification: 140. (**C**) RT-PCR/Sanger sequencing revealed the sequence of the novel EWSR1-PBX3 fusion transcript. Partial sequence chromatogram showing the junction (arrow) of the *PBX3* and *EWSR1* genes.

**Figure 5 ijms-24-00632-f005:**
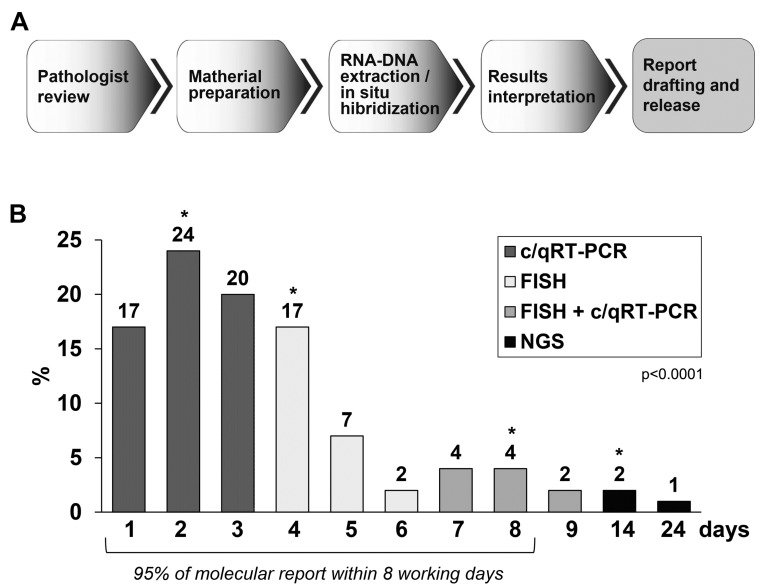
Workflow and turnaround time for clinical molecular samples (the year 2019–2020). (**A**) Workflow labelled with definitions of turnaround time metrics. (**B**) Histogram of the turnaround time from the start of tissue sample preparation for molecular investigation to report sign out (laboratory time) measured in calendar days. The grey bars represent the samples analysed with c/qRT-PCR, the white bars represent the samples analysed with FISH, the hatched bars represent the samples that required both c/qRT-PCR and FISH investigations, and the black bars represent the NGS analyses. Hatched and filled bars together add up to 100%. An Athena laboratory information system (Dedalus, Firenze, Italy) was used for specimen tracking. Cases were scanned to track their status as they progressed through the steps of the laboratory workflow. Time tracking data were extracted via a query). *p* value refers to the median differences in time (days) between the 4 groups. * median value of each group.

**Table 1 ijms-24-00632-t001:** Tumours tested for molecular diagnostic confirmation with total positive findings by c/qRT-PCR and FISH.

Tumour Tested (Suspected Cases Based on Histology and IHC)	n. Cases Tested	n. c/qRT-PCR Tested	n. FISH Tested	n. c/qRT-PCR Positive	n. FISH Positive	n. c/qRT-PCR Failed	n. FISH Failed
Alveolar rhabdomyosarcoma	47	30	25	23	6	6	7
Alveolar soft part sarcoma	17	14	5	14	2	1	0
Aneurismal Bone cyst	77	29	49	17	11	6	7
Angiomatoid fibrous Histiocytoma	8	6	5	4	2	2	0
Cartilage-forming tumours (*IDH1*, *IDH2*)	35	28	n.a.	17	n.a.	4	n.a.
*CIC*-rearranged sarcoma	35	21	25	18	13	6	2
Clear cell sarcoma	51	42	27	26	3	3	7
Chondroblastoma (*H3F3B*)	20	5	n.a.	9	n.a.	0	n.a.
Dermatofibrosarcoma protuberans	13	10	11	8	2	0	1
Desmoplastic small round cell tumour	9	6	6	5	2	0	0
Epithelioid Haemangioendothelioma	34	20	17	6	2	3	8
Epithelioid Haemangioma	10	9	2	2	0	1	n.a.
Ewing sarcoma	761	602	383	432	151	99	55
Extraskeletal Myxoid Chondrosarcoma	125	95	78	58	17	13	9
Giant cell tumour of bone (*H3F3A*)	71	71	n.a.	36	n.a.	2	n.a.
Infantile sarcoma	22	20	8	6	1	3	0
Low-grade central osteosarcoma (*MDM2)*	28	n.a.	28	n.a.	10	n.a.	5
Low-grade fibromyxoid sarcoma	9	8	1	8	1	0	0
Mesenchymal Chondrosarcoma	20	18	4	16	1	3	1
Mixoid liposarcoma	388	297	185	185	101	45	25
Molecular counselling	73	n.a.	n.a.	n.a.	n.a.	n.a.	n.a.
Myoepithelial tumour	13	5	13	0	3	5	1
Parosteal osteosarcoma (*MDM2*)	9	n.a.	9	n.a.	3	n.a.	2
Pseudomyogenic Haemangioendothelioma	3	3	n.a.	0	n.a.	3	n.a.
Round cell sarcoma with *EWSR1*-non ETS fusions	15	12	15	0	2	5	6
Sarcoma with *BCOR* genetic alterations	14	11	11	10	4	0	0
Sclerosing Epithelioid Fibrosarcoma	9	6	9	3	0	4	6
Synovial sarcoma	393	315	192	265	81	48	34
Well/dedifferentiated liposarcoma *MDM2* status	396	n.a.	396	n.a.	156	n.a.	61

Legend: n.a., not applicable. FISH, fluorescence in situ hybridization; c/qRT-PCR, conventional and quantitative transcription polymerase chain reaction; IHC immunohistochemistry.

**Table 2 ijms-24-00632-t002:** Sensitivity and specificity of the molecular analyses in the diagnosis of different subtypes of BSTS.

BSTS Subtype	Molecular Test	Sensitivity	Specificity
**Ewing sarcoma**	c/qRT-PCR [t(11;22) t(t21;22)]	94%	100%
	EWSR1 FISH	85%	94%
	Both test	99%	99%
**Synovial sarcoma**	c/qRT-PCR t(X;18)	95%	100%
	SS18 FISH	75%	100%
	Both test	98%	100%
**Mixoid liposarcoma**	c/qRT-PCR [t(12;16) t(t12;22)]	92%	100%
	DDIT3 FISH	99%	100%
	Both test	99%	100%
**Extraskeletal mixoid chondrosarcoma**	c/qRT-PCR [t(9;22) t(t15;22) t(17;22)]	88%	100%
	NR4A3 FISH	94%	100%
	Both test	97%	100%

BSTS indicates bone and soft tissue sarcoma; FISH, fluorescence in situ hybridization; c/qRT-PCR, conventional and quantitative transcription polymerase chain reaction.

## Data Availability

Data available on request due to restrictions, e.g., privacy or ethical. The data presented in this study are available on request from the corresponding author.

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
