# Peer review of "The Efficacy of Molecular Analysis in the Diagnosis of Bone and Soft Tissue Sarcoma: A 15-Year Mono-Institutional Study"

_ijms, 2022, doi:10.3390/ijms24010632_

Round 1

Reviewer 1 Report

Dear editor and authors, I have read and enjoyed this manuscript very much.

The language is adequate for a scientific publication.

The objectives have been fully achieved.

The results support the conclusions quite well.

The manuscript deserves publication. I found 2 critical issues that must be considered by the authors.

 (i) The manuscript is totally devoid of statistics. The authors show different estimates but without error. This aspect is fundamental. Also consider adding statistical tests where needed. I think this is very important.

(ii) In my opinion, the discussion (section 3) is slightly lacking in comparison with the literature. Authors may add comparisons and references. Few but incisive sentences.

Minor Issues

- It appears that this manuscript was previously submitted to another (i.e. Physica Medica EJMP – Elsevier, maybe). I ask the authors to review the organization of the abstract, and of the discussions in accordance with the journal.

- Read the entire manuscript carefully and correct various typing errors. Focus on tenses.

I congratulate the IT authors.

Author Response

Response to Editor and Reviewer comments:

 â€¯
REVIEWER 1
The manuscript deserves publication. I found 2 critical issues that must be considered by the authors.

(i) The manuscript is totally devoid of statistics. The authors show different estimates but without error. This aspect is fundamental. Also consider adding statistical tests where needed. I think this is very important.

According to reviewer’s comment, the authors reported statistical analysis where it was lacking (see Fig 3 and Fig 5). This work describes the molecular diagnostic activity performed over 15 years. In our opinion, supported by the consultancy with the statistical collaborator, it does not require further statistical analysis

(ii) In my opinion, the discussion (section 3) is slightly lacking in comparison with the literature. Authors may add comparisons and references. Few but incisive sentences.

According to reviewer’s comment the discussion (section 3) has been modified, partially rewritten with added comparisons and bibliographic references.

Minor Issues

- It appears that this manuscript was previously submitted to another (i.e. Physica Medica EJMP – Elsevier, maybe). I ask the authors to review the organization of the abstract, and of the discussions in accordance with the journal.

According to reviewer’s comment, we have revised the organization of the abstract and discussion in accordance with the journal. The abstract had been submitted for VIRCOWS ARCHIVES in November 2021; now the Content Files are Deleted (Forced to Withdrawn).

- Read the entire manuscript carefully and correct various typing errors. Focus on tenses.

According to reviewer’s comment, we have corrected the typing errors and the tenses.

Reviewer 2 Report

1.       The research covers the latest developments in the diagnosis of bone and soft tissue sarcoma.

2.       Some parts have been repeated many times, which should be reviewed and edited.

3.       The manuscript would benefit from English proofreading. Examples:

·         Results, line 77: Please change “In case” to “In the case”

·         Results, line 79: Please change: “Most of the PCR assays have been collected thanks to the available scientific literature” to “Most of the PCR assays have been used according to the available scientific literature”.

·         Table 1: Please use either an uppercase or a lowercase as the abbreviation for the number.

·         Results, line 108: Please change “In 14% the gene rearrangement analyses” to “In 14% of the gene rearrangement analyses”

·         Figure 1.: Please change “in 15 year of activity” to “in 15 years of activity” and “based on type of analysis” to “based on the type of analysis”

·         Results, line 134: Please change “16% were unsuitable material” to “in 16% material was unsuitable for analysis”.

·         Results, line 139: Please change “referrals samples” to “reference samples” and move the text in parentheses in front of “consultation cases”

·         Results, line 140: Please change “referrals” to “reference samples”

·         Table 2.: Please change “analisys” to “analysis” and rewrite Table 2. caption to better reflect the results, perhaps “Sensitivity and specificity of the molecular analyses in the diagnosis of different subtypes of BSTS” would be more appropriate.

·         Results, line 184: Please change: “To evaluate of the analytical quality” to “To evaluate the analytical quality”

·         Results, lines 188 and 189: Please rewrite the sentence to clarify.

·         Figure 3.: Please change “in traditional way” to “in a traditional way”.

·         Results, line 220: “In cases 4 and 15, gene fusions which are characteristic of ES were detected.” instead of “Cases 4 and 15 resulted in gene fusions characteristic of ES“ would be more appropriate.

·         Results, line 224: Please rewrite the sentence to clarify.

·         Authors should be consistent and use either “tumor” or “tumour” throughout the manuscript.

·         Figure 5.: Please change “year” to “the year”.

·         Discussion, line 314: Please delete “A”.

·         Discussion line 325: Please delete “the”.

·         Discussion, line 331: Please change “of bone tissue sarcomas” to “bone tissue sarcomas”

·         Discussion, line 344: Please change “the tests availability” to “the test’s availability”

·         Discussion, lines 363 and 364: Please rewrite the sentence to clarify.

·         Discussion, line 372: Perhaps it would be more appropriate to say: ”…were internal, from the Institute…”

·         Discussion, line 375: Please change “gene fusion” to “gene fusions”

·         Discussion, lines 411 and 412: It’s recommended to rewrite the sentence. Probably: “…has allowed us to improve the performance of molecular diagnostic tests.” would be better.

·         Please change “no evaluable” to “not evaluable” throughout the manuscript.

·         Authors should check punctuation throughout the manuscript, especially comma errors.

·         Discussion, line 433: Please change “detection” to “the detection”

·         Discussion, line 444: Please change “…the sorting out of driver mutations from passenger mutations…” to “…sorting out driver mutations from passenger mutations…”

·         Discussion, line 457: Perhaps: “…, development of a specific c/qRT–PCR assay and performing Sanger sequencing for confirmation” would be more appropriate.

·         Materials and Methods, lines 460 and 461: Please rewrite the sentence to clarify.

·         Materials and Methods, line 466: Please change “diagnostic second opinion” to “a second opinion on a diagnosis”

·         Materials and Methods, line 468: Please change “under-vacuum sealed” to “vacuum sealed”

·         Materials and Methods, line 471: Please change “Once in the pathology laboratory” to “Once in pathology laboratory”

·         Materials and Methods, line 474: Please use a more precise term instead of “molecular means”

·         Materials and Methods, line 480: Please change “of” to “for”.

·         Materials and Methods, line 485: Please change “case” to “the case”.

·         Please use “undiagnosed” instead of “unresolved” and “diagnosed” instead of “solved”.

·         Materials and Methods, line 500: Please change “To extract RNA frozen tissue” to “To extract RNA from frozen tissue”.

·         Materials and Methods, line 504: Please change “a no template control” to “no template control”.

·         Materials and Methods, line 529: “The histological analysis” instead of “the histology” would be more appropriate.

·         Materials and Methods, line 563: Please change “in according with” to “according to”.

·         Materials and Methods, line 574: Please change “Sample” to “The sample”.

·         Materials and Methods, lines 576, 577, 584 and 585: Please rewrite the sentences to clarify.

·         Materials and Methods, line 590: It’s recommended to add: “for comparison to obtained sequences” at the end of the sentence.

4.       The Authors should explain the term “strong fusion” and how it differs from a “fusion gene”.

5.       In the Conclusion section, it’s recommended to change “the highest sensitivity and specificity” to higher sensitivity and specificity” since some samples were negative by the first molecular screening (c/qRT–PCR and FISH) and positive after testing using NGS.

Author Response

REVIEWER 2

Comments and Suggestions for Authors

  1. The research covers the latest developments in the diagnosis of bone and soft tissue sarcoma.
  2. Some parts have been repeated many times, which should be reviewed and edited. According to reviewer’s suggestion, we have revised and modified the manuscript by removing the repeated part.
  3. The manuscript would benefit from English proofreading. Examples: According to reviewer’s comment, we have carefully revised the manuscript
  • Results, line 77: Please change “In case” to “In the case” Following the reviewer suggestions, we change “In case” to “In the case”
  • Results, line 79: Please change: “Most of the PCR assays have been collected thanks to the available scientific literature” to “Most of the PCR assays have been used according to the available scientific literature”. Following the reviewer suggestions, we changed it.
  • Table 1: Please use either an uppercase or a lowercase as the abbreviation for the number. Following the reviewer suggestions, we changed it.
  • Results, line 108: Please change “In 14% the gene rearrangement analyses” to “In 14% of the gene rearrangement analyses” Following the reviewer suggestions, we changed it.
  • Figure 1.: Please change “in 15 year of activity” to “in 15 years of activity” and “based on type of analysis” to “based on the type of analysis” Following the reviewer suggestions, we changed it.
  • Results, line 134: Please change “16% were unsuitable material” to “in 16% material was unsuitable for analysis”. Following the reviewer suggestions, we changed it.
  • Results, line 139: Please change “referrals samples” to “reference samples” and move the text in parentheses in front of “consultation cases” Following the reviewer suggestions, we changed it.
  • Results, line 140: Please change “referrals” to “reference samples” Following the reviewer suggestions, we changed it.
  • Table 2.: Please change “analisys” to “analysis” and rewrite Table 2. caption to better reflect the results, perhaps “Sensitivity and specificity of the molecular analyses in the diagnosis of different subtypes of BSTS” would be more appropriate. According to reviewer’s suggestion, the authors have revised the Table2. caption.
  • Results, line 184: Please change: “To evaluate of the analytical quality” to “To evaluate the analytical quality” Following the reviewer suggestions, we changed it.
  • Results, lines 188 and 189: Please rewrite the sentence to clarify. Following the reviewer suggestions, we have edited the sentence to clarify.
  • Figure 3.: Please change “in traditional way” to “in a traditional way”. Following the reviewer suggestions, we changed it.
  • Results, line 220: “In cases 4 and 15, gene fusions which are characteristic of ES were detected.” instead of “Cases 4 and 15 resulted in gene fusions characteristic of ES“ would be more appropriate. Following the reviewer suggestions, we changed it.
  • Results, line 224: Please rewrite the sentence to clarify. Following the reviewer suggestions, we have edited the sentence to clarify.
  • Authors should be consistent and use either “tumor” or “tumour” throughout the manuscript. Following the reviewer suggestions, we have consistently chosen to use “tumour” throughout the manuscript
  • Figure 5.: Please change “year” to “the year”. Following the reviewer suggestions, we changed it.
  • Discussion, line 314: Please delete “A”. Following the reviewer suggestions, we deleted it.
  • Discussion line 325: Please delete “the”. Following the reviewer suggestions, we deleted it.
  • Discussion, line 331: Please change “of bone tissue sarcomas” to “bone tissue sarcomas” Following the reviewer suggestions, we changed it.
  • Discussion, line 344: Please change “the tests availability” to “the test’s availability” Following the reviewer suggestions, we changed it.
  • Discussion, lines 363 and 364: Please rewrite the sentence to clarify. Following the reviewer suggestions, we have edited the sentence to clarify.
  • Discussion, line 372: Perhaps it would be more appropriate to say: ”…were internal, from the Institute…” Following the reviewer suggestions, we changed it.
  • Discussion, line 375: Please change “gene fusion” to “gene fusions” Following the reviewer suggestions, we changed it.
  • Discussion, lines 411 and 412: It’s recommended to rewrite the sentence. Probably: “…has allowed us to improve the performance of molecular diagnostic tests.” would be better. Following the reviewer suggestions, we rewrote the sentence.
  • Please change “no evaluable” to “not evaluable” throughout the manuscript. Following the reviewer suggestions, we changed it.
  • Authors should check punctuation throughout the manuscript, especially comma errors. According to reviewer’s comment, we have checked the punctuation.
  • Discussion, line 433: Please change “detection” to “the detection” Following the reviewer suggestions, we changed it.
  • Discussion, line 444: Please change “…the sorting out of driver mutations from passenger mutations…” to “…sorting out driver mutations from passenger mutations…” Following the reviewer suggestions, we changed it.
  • Discussion, line 457: Perhaps: “…, development of a specific c/qRT–PCR assay and performing Sanger sequencing for confirmation” would be more appropriate. Following the reviewer suggestions, we changed it.
  • Materials and Methods, lines 460 and 461: Please rewrite the sentence to clarify. Following the reviewer suggestions, we have edited the sentence to clarify.
  • Materials and Methods, line 466: Please change “diagnostic second opinion” to “a second opinion on a diagnosis”. Following the reviewer suggestions, we changed it.
  • Materials and Methods, line 468: Please change “under-vacuum sealed” to “vacuum sealed”. Following the reviewer suggestions, we changed it.
  • Materials and Methods, line 471: Please change “Once in the pathology laboratory” to “Once in pathology laboratory”. Following the reviewer suggestions, we changed it.
  • Materials and Methods, line 474: Please use a more precise term instead of “molecular means”. Following the reviewer suggestions, we changed it.
  • Materials and Methods, line 480: Please change “of” to “for”. Following the reviewer suggestions, we changed it.
  • Materials and Methods, line 485: Please change “case” to “the case”. Following the reviewer suggestions, we changed it.
  • Please use “undiagnosed” instead of “unresolved” and “diagnosed” instead of “solved”. Following the reviewer suggestions, we changed it.
  • Materials and Methods, line 500: Please change “To extract RNA frozen tissue” to “To extract RNA from frozen tissue”. Following the reviewer suggestions, we changed it.
  • Materials and Methods, line 504: Please change “a no template control” to “no template control”. Following the reviewer suggestions, we changed it.
  • Materials and Methods, line 529: “The histological analysis” instead of “the histology” would be more appropriate. Following the reviewer suggestions, we changed it.
  • Materials and Methods, line 563: Please change “in according with” to “according to”. Following the reviewer suggestions, we changed it.
  • Materials and Methods, line 574: Please change “Sample” to “The sample”. Following the reviewer suggestions, we changed it.
  • Materials and Methods, lines 576, 577, 584 and 585: Please rewrite the sentences to clarify. Following the reviewer suggestions, we have edited the sentence to clarify.
  • Materials and Methods, line 590: It’s recommended to add: “for comparison to obtained sequences” at the end of the sentence. Following the reviewer suggestions, we have added it.
  1. The Authors should explain the term “strong fusion” and how it differs from a “fusion gene”. The authors chose to use the term “strong fusion” because the Archer’s software analysis suggests this terminology when it finds the presence of a gene fusion. Following the reviewer’s suggestion, we decided to replace the term “strong fusion” with “fusion” because it could be confusing.
  2. In the Conclusion section, it’s recommended to change “the highest sensitivity and specificity” to ”higher sensitivity and specificity” since some samples were negative by the first molecular screening (c/qRT–PCR and FISH) and positive after testing using NGS. Following the reviewer suggestions, we changed it.

Round 2

Reviewer 1 Report

The Authors have carefully revised their paper according to all comments and given response to the reviewers separately. Therefore, manuscript can be accepted for publication.